# Inferences from Portfolio Theory and Efficient Market Hypothesis to the Impact of Social Media on Sovereign Debt: Colombia, Ecuador, and Peru

Esteban Serrano-Monge 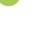

Business School, Universidad San Francisco de Quito-USFQ, Diego de Robles y Vía Interoceánica, Quito 170157, Ecuador; eserrano@usfq.edu.ec; Tel.: +593-2-297-1700

**Abstract:** For three countries of similar economic characteristics, I ratify previous studies of the impact of fundamental macroeconomic and foreign exchange variables influencing country risk, as captured by the Emerging Market Bond Index (EMBI). I contribute to existing research, first by calculating a proxy of risk I call endogenous risk that analyzes the quarterly variability of economic activity, and second, by calculating a variable of sentiment from Twitter activity. I gauge the impact of both on the country risk metric in addition to variables in existing research about the determinants of country risk. Foreign exchange variables are the most significant determinants of risk for the countries of Colombia and Peru, which actively manage their currency, while Ecuador's country risk is mostly affected by endogenous risk and macroeconomic fundamentals.

**Keywords:** efficient market hypotheses; portfolio theory; sovereign credit risk; country risk; social media

## 1. Introduction

This research focuses on the study of the causal relationships revealed by the analysis of macroeconomic fundamentals, sovereign bonds trading activity, and news sentiment as predictors of JP Morgan's Emerging Markets Bond Index (EMBI) spread for the countries of Colombia, Ecuador, and Peru. The risk of a country's sovereign bond has extensively been proxied by the literature as the spread or difference between the return of a specific country's EBMI versus the return of an equivalent maturity US Treasury security.

The time frame selected for this research (2000–2019) is the period following Ecuador's adoption of the US Dollar as its official currency (Mahuad 2021), an event that resulted in an ad hoc experimental setting conducive to the research of these three geographically neighboring, but more importantly, commodity-dependent countries (Ocampo 2017). Furthermore, the exchange rate policies of these three countries also provide an appealing contrast, ranging from non-existent (Ecuador) to managed exchange rate and inflation targeting policies as exhibited by Colombia and Peru (Libman 2019).

The yield and return of the sovereign bonds which comprise the EMBI are the results of the periodical variations in the marked-to-market pricing of these bonds, whereby the holding period return is affected by two concurrent factors. On the one hand, the expectations of default based on the actual and perceived track record of a country's overall economic fundamentals typically inform an investor via the ratings released by entities that analyze the sovereign risk of a specific issue (bond) and issuer (country). In addition, a bond's yield is also the result of transactions and regular trading activity in the capital markets motivated by factors such as the liquidity of a security, daily foreign exchange rates affecting the issuer's currency, and any information which percolates to the news media about the country, all of which is commonly referred to as market sentiment (JP Morgan 1999).

As classified by Mari Del Cristo and Gómez-Puig (2017), research attempting to identify the causes of country risk fall under three categories. In the first category, macroeconomic and political variables act as determinants of country risk, the second category emphasizes exogenous factors (such as market or investor sentiment, contagion effects, capital flows) as determinants, and the third and final group emphasize exchange rate regime as a determinant of country risk.

I performed OLS regressions of 14 independent variables covering these three categories of research against three variants of the dependent EMBI spread. From the literature reviewed, there appear to be three independent variables that constitute a differentiated contribution in the study of country risk determinants: endogenous risk and the impact of Twitter ® activity from both regular and financial news outlets (described in Section 3). The literature provides extensive research about market sentiment; according to Gan et al. (2020), these studies can be categorized by the sentiment measure they use. Namely, measures of fundamental market variables extracted from textual sources and sentiment scores extracted from proprietary vendors such as Thomson Reuters MarketPsych® and RavenPack®.

In the first category of research, results for the three countries studied confirm the findings of the causal relationships between the level of indebtedness (positively affecting risk premium) and reserves (negatively affecting risk premium) to the EMBI spread, as revealed by Edwards (1986). Being a variability measure of macroeconomic activity, I place the posited endogenous risk variable in this category, and it is only of relative significance for Ecuador, positively affecting its level of country risk.

In the second category of exogenous factors, the results are mixed. Corruption levels positively affect Colombia's risk premium. Conversely, a positive business environment, not surprisingly, negatively affects its country's risk level. Peru's spreads are influenced by the liquidity of its traded debt issue.

Under the third category of exchange rate regime effects, the variable capturing the standard deviation of the foreign exchange rate positively affects the EMBI spread of Peru and Colombia while, as expected, having no relevance to Ecuador's country risk level.

I set out to investigate hypotheses 1–5 described below. Hypothesis 2, 3, and 4 capture variables in the above-referenced research categories, while Hypothesis 1 (endogenous risk) and 5 (Twitter ® activity) capture the novel variables posited.

### 1.1. Hypotheses

**Hypothesis 1 (Endogenous Risk).** *The results of the research were anticipated to reject the null hypothesis ($H_0$) that endogenous risk contributed no statistically significant predictive ability in the determination of either variant of the EMBI spread for the countries studied.*

**Hypothesis 2.** *A hypothesis test was also conducted concerning the quantitative, survey- and perception-based macro variables, STARTING A BUSINESS and CORRUPTION. I anticipated a weak predictive ability from the latter, yet sufficient to reject the null hypothesis $H_0$ that both variables contributed no statistically significant predictive ability in the determination of the EMBI spread for the three countries studied.*

**Hypothesis 3.** *Positive hypotheses were also formed with respect to capital market attributes such as liquidity, captured by the bid–ask spread, AVERAGE PERIOD RATING captured by a credit-rating score, the changes in value of local currency (only applicable to Colombia and Peru, since Ecuador adopted the US dollar as its official currency at the onset of 2000), FX DepR_AppR, and the standard deviation of changes of these exchange rates STD. DEV of F/X. The more flexible versions of the credit ratings and bid–ask spread metrics, characterized in the form of binary qualitative variables, investment grade or not AVG. invest. grd. (1) or not (0) and traded or not TRADED = 1 NON-TRADED = 0, were hypothesized in the same light.*

**Hypothesis 4.** *This hypothesis reiterates the impact from the macroeconomic indicators (e.g., FDI/GDP, Debt Svce./GDP, Reserves/GDP) extensively identified as significant in research about the determinants of country risk.*

**Hypothesis 5 (Twitter ® Activity).** *I anticipated the results to accept the null hypothesis $H_0$ since both ALL Tweet and FIN Tweet were deemed to have limited explanatory powers to determine the variants of the EMBI spread.*

A description of the variables that contribute to this research is detailed below:

*1.2. Twitter Impact*

The simple extraction of activity on the social media platform Twitter was not enough since the operationalization of this variable, adopted to try and ascertain the impact of reputation on a country's EMBI spread, needed refinement to be regressed against the observed variant of the dependent variable. For this reason, tweets were first divided by origin of media, between those coming from financial media (FIN tweets, see below) and those from general news media (ALL tweets, see below).

The underlying rationale used is that a larger captive audience would consequently create a greater impact on reputation. Whether this impact was negative, positive, or non-existent was not relevant to operationalizing this variable since the regression results would shed light on the direction of that effect.

The operationalized variable was arrived at by weighing the number of likes and retweets and adding this figure into the numerator, and later dividing it by the number of followers of the media outlet at the moment and time of a particular tweet containing the hashtag of the country's name. The weights of a like and retweet were determined by prospecting research work by Meier et al. (2014) concerning the behavior of users of social media and the level of importance given to both activities. The reasoning from this review is such that a retweet is given twice as much importance as a like in terms of the impact it bears on sentiment. Finally, this proportion (# of followers) was summed for the period studied (one trimester) to coincide with the quarterly economic-activity data of the country and the observed EMBI spread.

*1.3. Endogenous Risk*

Two options were considered to operationalize the endogenous risk variable. First, one could utilize the value of capital in place, which is the result of capital formation or build-up for each individual industry, to discover the return provided by those assets. This alternative was discarded for lack of comparable information and the complexity involved in determining the net return of each industrial activity across countries. Instead, the value-added factor of economic activity, classified with an international standard industrial classification (ISIC) structure, provided a comparable basis from which to calculate the variability of an economy resulting from the interactions of each industry within it.

The following steps were taken to operationalize the endogenous-risk independent variable:

1.  For each country studied, the longest available quarterly time series of industrial economic value-added activity was retrieved.
2.  From the above information, the quarterly rate of change of each industrial activity was calculated. This calculation was analogous to calculating the rate of return from a time series of security prices.
3.  Consequently, covariance and correlation matrices of these rates of change were calculated.
4.  Each quarterly industrial economic activity was weighted to represent its proportion to the total economic activity. This calculation was analogous to calculating the proportion of a financial asset to the total assets in a portfolio of the same.

5.  By utilizing the covariance matrices, the contribution of each individual industrial economic activity's quarterly variance of returns to the total variance of the whole of the economy was calculated based on its respective weight.
6.  The square root of the variance for each quarter resulted in the standard deviation or endogenous risk of each quarterly return for the economy, and this was the observable independent variable used in the regression against the variations of the EMBI spread time series.

The rest of this paper is organized as follows. Section 2 provides an overview of the pertinent literature. Section 3 describes the data and variables, and process. Section 4 provides discussion and conclusions.

## 2. Literature Overview

### 2.1. Sovereign Credit Risk

Sovereign, or government, debts are significantly older financial instruments than other high-risk assets, such as stocks. In his book *The Ascent of Money*, Niall Ferguson (2008) pegs the birth of the debt bond market during the Italian Renaissance period, where sovereign debt was used to finance the defense or expansion of territories, governments, and kingdoms in the mid-fourteenth century. Long-term public debt first emerged in Europe's autonomous smaller cities rather than in the larger territories. Tuscan city-states such as Florence, Pisa, and Siena financed their wars through loans from their inhabitants. City states were considered more creditworthy than larger territorial states and were thus able to access a lower interest rate. The Dutch Republic was the first territorial state which was able to obtain similar conditions to the smaller city states (Stasavage 2016).

Since the inception of debt instruments, the causality behind differing borrowing rates for distinct sovereignties has been empirically analyzed in an attempt to establish the attributes which determine the varying conditions faced by sovereign borrowers. For example, Dincecco (2009) suggested that the type of political regime, fiscal management, as well as cultural attitudes towards the responsibilities of an indebted country or territory were determinant variables for the probability of default. At the cusp of the modern era of finance, defined as the period after the gold standard was abandoned (Eatwell and Taylor 1998), empirical quantitative analyses of macroeconomic variables and indices have been conducted to establish the underlying causes of sovereign risk and their probability.

Since the mid-1970s, there has been a significant expansion of research concerning sovereign debts and the variables that determine their probability of default for the countries defined as emerging markets and, even more specifically, those in Latin America. These countries have been estimated to account for 71% of the volume of debt instruments that constitute the EMBI (JP Morgan 2017), an aspect that undoubtedly attracted the interest of authors such as Edwards (1984) and Grandes (2007), with close ties to the Latin American region.

In the 1970s, the rapid rise of external debt obligations of developing countries encouraged researchers to analyze their risks of default. The most visible works in this area came from a professor at the University of California Los Angeles (UCLA), Sebastián Edwards (1984, 1986), as well as Sachs (1988); McDonald (1982); Eaton and Gersovitz (1980); and Feder and Just (1977). These researchers analyzed multiple regression models, postulating time series of macroeconomic indicators as determinants of the differential, or spread between the base rates (at that time, the London Inter-Bank Offered Rate (LIBOR) rate; today, this spread is calculated between the yield on the sovereign bond and the current issuance of a North American Treasury bond) as independent variables of the model and the debt rates of the lesser-developed country (LDC) as the dependent variables. The authors Homi Kharas (1984) and Kamin and von Kleist (1999) analyzed the risk of the debts of the lesser-developed, developing, and emerging countries (depending on the time frame studied, these were the terms used) from the mid-to-late 1980s and until the end of the last century. In the latter, the authors separated the effects of the Brady bonds and included other types of debts in countries that were not considered developed economies.

At the same time, Goldman Sachs (2000) produced a model dubbed the Goldman Sachs Equilibrium Sovereign Spread (GS-ESS), which built upon the elements studied by Edwards (1986) to explain the yield spread. This model grouped macroeconomic variables and indices and thus simplified the work of Edwards (1986). Martín Grandes (2007) conducted an analysis of historical prospecting about the determinants of yield spreads in Latin America based on empirical evidence from Mexico, Brazil, and Argentina.

Academic studies that examine macroeconomic relationships as independent variables to predict the risk of default for a particular country abound (Cosset and Roy 1991; Vij 2005; San-Martín-Albizuri and Rodríguez-Castellanos 2011). As such, the analysis behind the decision to invest in a particular sovereign debt instrument can be supported by thorough research regarding the underlying causes and risks of default.

Additional research by Edwards (1986), studies by the investment banking industry (Goldman Sachs 2000), central banks such as Banco de la República-Colombia (Rowland and Torres 2004), and multilateral financial entities such as the World Bank Development Research Group (Min 1999) and the Bank for International Settlements (Kamin and von Kleist 1999) have delved into the determinants of fair value and default risk probability for sovereign debt instruments.

The research probed into the causes of fair value and default risk by conducting series analyses of fundamental macroeconomic metrics, such as reserves to gross domestic product (GDP) and debt to GDP ratios, and on capital market variables, such as liquidity or credit ratings. These series were then regressed against the actual EMBI spread as the dependent variable to determine their predictive ability.

The work by Rowland and Torres (2004) deftly described these types of studies. The body of research in this area (e.g., sovereign risk, probability of default, and spread to a benchmark rate) has used economic data, political and social factors, as well as market sentiment and contagion factors, to measure the sovereign risk of a particular debt issue, as detailed in Figure 1. I used the referenced framework (Rowland and Torres 2004) as a building block to explore additional variables within the context of the determinants of sovereign debt risk.

### 2.2. Market Sentiment and Contagion

The efficient market hypothesis (Fama 1970), EMH for short, posits that market efficiency should be measured from three perspectives: the most basic form (called the "weak" form) states that the past behavior of securities' prices should fully inform the current prices of those securities. The second variant, referred to as the "semi-strong" form, gauges the current impact of newly available public information on the securities' prices. Lastly, the third, "strong" form tests whether private, non-public information has been reflected in the market prices.

Of significant interest to this research is the question of whether publicly available information broadcast through Twitter has affected the prices of sovereign bond securities or if this information was pure noise and consequently had neither a real impact on the prices of relevant securities nor a predictive ability to forecast price and return variables. The impact of information on equity securities has been widely studied; however, the impact of social media on sovereign credit risk has been largely overlooked (Erlwein-Sayer 2018).

The impact was measured by weighing the number of followers of a specific news outlet via their Twitter handle during a particular bi-weekly period and computing a simple sum of the impact of "tweets", including published statements, commentary, etc., by the number of "likes" (i.e., an indicator of recognition, support, etc. from other Twitter users in response to a tweet) and "retweets" (i.e., republishing the original tweet by other Twitter users with or without their own commentary attached) in response to the original tweet. Succinctly, the results of the model suggest that new publicly available information affected the prices and the yields of sovereign bonds before the same information was released via

a tweet or retweet and thus rendered the social media impact ineffective since the news was revealed to the public via another medium, had already affected prices and yields.

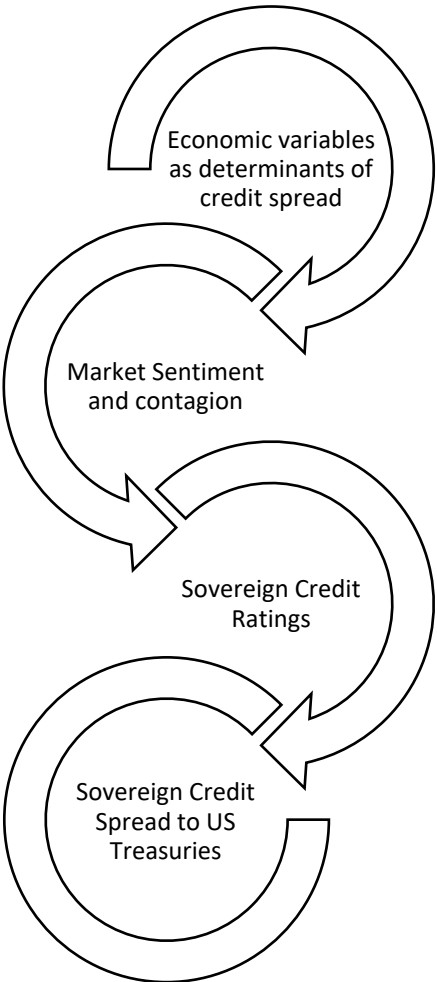

**Figure 1.** Adapted from Theoretical Frameworks of Rowland and Torres (2004). Created by the author.

My research involved gathering empirical data to shed light on the possible economic relationships to predict an outcome from these observed empirical datasets, or what Friedman (Friedman and Friedman 1953) called "positive economics". A research model is designed to represent what is, not what ought to be, from a dogmatic perspective; however, the regression models I used could only capture a portion of the true relationships that inform the prices of securities, which is true of any regression model.

The hypotheses were designed to discard the effect that new information published via a social media outlet, namely Twitter [®], would be followed by a change in prices and yields of sovereign securities. The precision of the model, gauged by its ability as an unbiased predictor of the dependent variable (in this case, the yields of sovereign bond issues), was an abstraction of the real relationships between these variables. As such, fine-tuning the operationalization of the data could result in the same independent variables becoming better predictors of the sovereign yield spread. This fine-tuning was not performed, but changing the timeframe of measurable impact (i.e., shifting the biweekly measurement of impact to a different time period) could have positively affected the predictive ability of the model. This measurement of impact is also referred to as the granularity of data in the literature about the impact of sentiment on securities prices (Gan et al. 2020).

Nonetheless, as Fama (1991) related in his follow-up report "Efficient Capital Markets II", market efficiency and the transmission of news (via Twitter and its tweets, retweets,

and likes, for this research) to prices cannot be tested in a vacuum. There must be some model of equilibrium, such as an asset-pricing model, to jointly test the EMH. In this study, I utilized the spread to the US Treasury (referred to as "country risk" in financial jargon) as the basis to identify the true equilibrium price (in the case of the sovereign bond, the yield spread) of the sovereign bond security. The question thus remained: if the model and the variables selected could not provide the desired predictability, was that due to the spread to the US Treasury being a poor gauge of relative risk, or were the model's independent variables at fault?

## 3. The Data

The data processed followed the sequence detailed below:

1.  Daily EMBI Spreads for each country were retrieved from January 2000 until December 2019.
2.  Three variants of this dependent variable were characterized as depicted in Section 3.3
3.  Unit Root (ADF) tests were performed on the time series of the dependent variables to test for non-stationarity and used the average spread of the EMBI for its stationarity (Table 1).
4.  I performed OLS regressions of Average EMBI Spread against the 14 independent variables described below
5.  Multicollinearity tests eliminated independent variables, and the surviving variables were analyzed for significance to test the hypotheses (Table 2).

**Table 1.** ADF test for non-stationary time series Average EMBI Spread.

| Country | Tau-Statistic | Tau-Critical Value | Stationary | AIC | BIC | LAGS | Coeff. | *p*-Value |
|---|---|---|---|---|---|---|---|---|
| Colombia | −3.76 | −3.48 | yes | 10.78 | 10.93 | 1 | −0.35 | 0.025 |
| Ecuador | −6.40 | −3.46 | yes | 14.69 | 14.81 | 1 | −0.43 | <0.01 |
| Peru | −4.16 | −3.49 | yes | 10.30 | 10.46 | 1 | −0.40 | <0.01 |

**Table 2.** Significant variables and Hypotheses test against Average EMBI Spread.

| Source | t | Pr > \|t\| | Accept H$_0$ | Reject H$_0$ |
|---|---|---|---|---|
| Colombia FX DepR_AppR | 4.647 | <0.0001 | | Hypothesis 3 |
| STD. DEV of F/X | 7.083 | <0.0001 | | Hypothesis 3 |
| Ecuador Endogenous Risk | −3.294 | 0.002 | | Hypothesis 1 |
| Ecuador Debt Svce./GDP | 2.979 | 0.005 | | Hypothesis 4 |
| Peru STD. DEV of F/X | 3.070 | 0.005 | | Hypothesis 3 |

### 3.1. Dependent (Y) Variables

1.  EMBI Spread—EMBI: observed at the end of the quarterly period to coincide with the series data for economic activity.
2.  AVG. Spread—EMBI: average for the observed quarterly period to coincide with the series data for economic activity.
3.  Std Dev Spread—EMBI: Standard deviation for the period to coincide with the series data for economic activity.

### 3.2. Independent (X) Quantitative Variables

1.  **All Tweets:** Twitter activity in the form of tweets was chosen as a research variable for this study as it has been one of the most widespread and internationally disseminated social media platforms (Pastel 2019). This social platform began activity in July 2006 (Arrington 2006), and approximately a year thereafter, hashtags were ubiquitously adopted. Information was extracted using a Twitter application programming interface (API) through a third-party application named www.trackmyhashtag.com and by searching the hashtags (#) Colombia, Ecuador, and Peru (Algodom Media LLP 2020). In so doing, Twitter activity from prominent media sources mentioning the countries'

names was extracted to determine if reputation, as a form of momentum, had affected sovereign security prices and, consequently, the EMBI spread of a particular country. An example of the information activity extracted is exhibited in Table 3.

**Table 3.** Twitter Activity for #Ecuador from 2007–2019 *.

| Name of Media | Handle | Retweets | Likes |
|---|---|---|---|
| BILD | @BILD | 170 | 225 |
| Bloomberg | @business | 6105 | 5879 |
| Chicago Tribune | @chicagotribune | 613 | 333 |
| Daily Mirror | @DailyMirror | 378 | 252 |
| EL PAÍS | @el_pais | 53,124 | 39,322 |
| Financial Times | @FinancialTimes | 2223 | 2224 |
| Washington Post | @washingtonpost | 6152 | 8460 |
| TIME | @TIME | 4768 | 3471 |
| USA TODAY | @USATODAY | 2259 | 1628 |

* Twitter, Inc., 1355 Market Street, Suite 900, San Francisco, CA 94103 U.S.A. Twitter application programming interface (API). Extracted with third-party application www.trackmyhashtag.com accessed on 14 June 2020.

2.  **FIN Tweets**: Twitter activity extracted following the same procedure used for the ALL Tweets variable (above), but from financial news media outlets active on Twitter. Specifically, *Bloomberg*, the *Financial Times*, the Economist Intelligence Unit, and *The Wall Street Journal*.

3.  **Endogenous Risk**: The intuition applied to operationalize this variable was to treat economic activities by industrial segments as if they were individual assets comprising a portfolio of assets, thereby applying the mean-variance analysis (Markowitz 1952).

4.  **STARTING A BUSINESS**: This is the yearly index calculated by the World Bank. "Distance to frontier score illustrates the distance of an economy to the 'frontier', which represents the best performance observed on each Doing Business topic across all economies and years included since 2005. An economy's distance to frontier is indicated on a scale from 0 to 100, where 0 represents the lowest performance and 100 the frontier. For example, a score of 75 in 2012 means an economy was 25 percentage points away from the frontier constructed from the best performances across all economies and across time. A score of 80 in 2013 would indicate the economy is improving," according to the World Bank Data Catalog (WBDC 2020).

5.  **CORRUPTION**: This variable was a yearly relative measure of corruption and represented a compilation retrieved from Transparency International's Corruption Perception Index (CPI) for the years that coincided with the EMBI spread period of analysis. The index calculation methodology changed in 2012; consequently, to make the time series comparable throughout the period of analysis, a percentile rank calculation was performed that computed a relative value of corruption perception for each one of the three countries studied in this research (Transparency International 2018).

6.  **FDI/GDP**: Foreign direct investment (FDI, net) as a percentage of GDP. GDP is the quarterly data annualized (i.e., multiplied times four). FDI data retrieved from the World Bank Data Catalog (WBDC 2020) coincided with the EMBI spread time series. "Foreign direct investment are the net inflows of investment to acquire a lasting management interest (ten percent or more of voting stock) in an enterprise operating in an economy other than that of the investor. It is the sum of equity capital, reinvestment of earnings, other long-term capital, and short-term capital as shown in the balance of payments. This series shows total net FDI. In BPM6, financial account balances are calculated as the change in assets minus the change in liabilities. Net FDI outflows are assets and net FDI inflows are liabilities. Data are in current U.S. dollars" (WBDC 2020). This variable was utilized in the regression with a negative sign to characterize the concept that a net FDI outflow was an asset and a net FDI inflow was a liability.

7.  **Debt Svce./GDP**: Debt service for all public debt as a percentage of GDP (WBDC 2020). Debt service definition: "Public and publicly guaranteed debt service is the sum

of principal repayments and interest actually paid in currency, goods, or services on long-term obligations of public debtors and long-term private obligations guaranteed by a public entity. Data are in current U.S. dollars" (WBDC 2020).

8.  **Reserves/GDP**: Reserves as a percentage of GDP (WBDC 2020). "Reserves and related items is the net change in a country's holdings of international reserves resulting from transactions on the current, capital, and financial accounts. Reserve assets are those external assets that are readily available to and controlled by monetary authorities for meeting balance of payments financing needs, and include holdings of monetary gold, special drawing rights (SDRs), reserve position in the International Monetary Fund (IMF), and other reserve assets. Also included are net credit and loans from the IMF (excluding reserve position) and total exceptional financing. Data are in current U.S. dollars" (WBDC 2020).

9.  **AVERAGE PERIOD RATING**: Rating refers to the average credit rating within a quarterly period to coincide with the EMBI spread time series. A compilation of ratings and a numerical score based on Bustillo et al. (2019) was computed (Table 4).

**Table 4.** Letter credit ratings with numerical score.

| S&P | Moody's | Fitch | Score |
| --- | --- | --- | --- |
| AAA | Aaa | AAA | 22 |
| AA+ | Aa1 | AA+ | 21 |
| AA | Aa2 | AA | 20 |
| AA− | Aa3 | AA− | 19 |
| A+ | A1 | A+ | 18 |
| A | A2 | A | 17 |
| A− | A3 | A− | 16 |
| BBB+ | Baa1 | BBB+ | 15 |
| BBB | Baa2 | BBB | 14 |
| BBB− | Baa3 | BBB− | 13 |
| BB+ | Ba1 | BB+ | 12 |
| BB | Ba2 | BB | 11 |
| BB− | Ba3 | BB− | 10 |
| B+ | B1 | B+ | 9 |
| B | B2 | B | 8 |
| B− | B3 | B− | 7 |
| CCC+ | Caa1 | CCC+ | 6 |
| CCC | Caa2 | CCC | 5 |
| CCC− | Caa3 | CCC− | 4 |
| CC | Ca | CC | 3 |
| C | C | C | 2 |
| SD | | RD | 1 |
| D | | D | 0 |

10. **Bid–Ask Spread (Liquidity):** The price differential between the ask and the bid, known as the bid–ask spread, has widely been used by the investment industry as a proxy for liquidity. This measure refers to the amount for which a security is bought and sold in the capital markets. The supposition was that the greater the liquidity, the less of a return an investor would demand and, thus, a lower cost and lower EMBI spread. The shortest-date issue (i.e., the issue with the longest available history) for each country analyzed in this study was selected as a proxy for liquidity, and the difference between the ask price and the bid price was calculated with reference to the ask price. Then, the quarterly daily-average bid–ask spread was calculated to coincide with the quarterly economic activity (Bloomberg Finance L.P. 2020a).

11. **Foreign exchange rate depreciation or appreciation (FX DepR_AppR)**: This variable calculates the appreciation or the depreciation of the currency measured against the US dollar to coincide with the quarterly observations of macroeconomic data. Data for

the foreign exchange rate quotes were extracted from Bloomberg (Bloomberg Finance L.P. 2020b).

12. **Standard deviation of foreign exchange rate (STD. DEV. of F/X)**: This variable depicts the standard deviation of the foreign exchange in direct quote terms (local currency/USD) corresponding to the quarterly time series of macroeconomic observations.

### 3.3. Independent (X) Qualitative (Dummy) Variables

13. **TRADED vs. NON-TRADED**: In the absence of a bid–ask spread, this variable was included to reflect the qualitative nature of a traded or non-traded security, whereas the magnitude of the bid–ask spread would hypothetically determine the magnitude of the EMBI spread, the hypothesis, when tested with this variable, was able to identify a simpler, binary concept that a non-traded security would result in a larger EMBI spread than the alternative, a security with trading activity.

14. **Investment Grade vs. Non-Investment Grade**: Similar to the previous qualitative variable, this was also a simpler version of the average-period-rating quantitative variable. In the former, the hypothesis could determine the impact of the average period rating by attaching a numerical score to the credit rating of a sovereign issue, whereas with this latter variable, the simpler notion of the rating being either investment grade or not was used to gauge its predictive ability in determining the EMBI spread of a sovereign issue.

A summary of the dependent and independent variables, the corresponding number of observations, and the regressions produced and studied are depicted in Table 5 below.

**Table 5.** General overview of empirical data analyzed.

| | Expected Relationship to Dependents | Colombia | Ecuador [1] | Peru |
|---|---|---|---|---|
| Number of raw observations of EMBI spread | | 5218 | 5218 | 5218 |
| Number of observations of EMBI spread with correspondence to independent variables | | 56 | 76 | 48 |
| Dependent variables of EMBIG spread (variants) [2] | | 3 | 3 | 3 |
| Number of observations for independent variable before series elimination: | | | | |
| 1. Endogenous Risk | + | 780 | 1501 | 728 |
| 2. STARTING A BUSINESS | − | 56 | 76 | 48 |
| 3. CORRUPTION | − | 56 | 76 | 48 |
| 4. FDI/GDP | − | 56 | 76 | 48 |
| 5. Debt Svce./GDP | + | 56 | 76 | 48 |
| 6. Reserves/GDP | − | 56 | 76 | 48 |
| 7. FX DepR_AppR | +/− | 5063 | n.a. | 5025 |
| 8. STD. DEV of F/X | + | 5063 | n.a. | 5025 |
| 9. ALL Tweets | +/- | 2738 | 1885 | 2747 |
| 10. FIN Tweets | +/− | 1020 | 365 | 654 |
| 11. Bid–Ask Spread (Liquidity) | + | 2321 | 1551 | 3880 |
| 12. AVERAGE PERIOD RATING | − | 50 | 67 | 44 |
| 13. TRADED = 1 NON-TRADED = 0 | +/− | 2321 | 1551 | 3880 |
| 14. AVG. Invest. Grd. (1) Or Not (0) | +/− | 50 | 67 | 44 |
| Number or Regressions Produced | | 13 | 18 | 15 |

[1] Ecuador's currency is the US dollar; thus, F/X variables were not applicable. [2] Three variants were studied: average (trimester) quarterly spread, end-of-quarter spread, and standard deviation of the spread over the quarter.

### 3.4. Empirical Findings from Regression Analysis

All of the independent variables used for this investigation were initially regressed against the three variations of the dependent variables to correspond with the series of

quarterly observations of macroeconomic activity, hereafter referred to as the complete regressions (CRs).

First, however, separate regressions were conducted for all instances where the independent variables retrieved contained more observations than the quarterly observations of macroeconomic activity. These separate regressions were performed to further confirm the relationships determined from the CRs since there were concerns regarding the possibility that restricting the number of observations from certain variables due to their lack of correspondence to the quarterly macroeconomic variable would handicap the predictive ability of the variable in question.

In so doing, these regressions gauged the unique predictive ability of the larger dataset of the independent variables against the also larger available dataset of the dependent EMBI spread. For instance, the ALL tweets, FIN tweets, and bid–ask spread variables were separately regressed against the three variants of the dependent EMBI spread variable. The resulting calculation for these series concluded in a weak explanatory power ($R^2$ less than 0.06 for all three countries) to predict the spread (Table 6).

**Table 6.** Larger Datasets *.

|  | ALL Tweet vs. AVERAGE EMBI Spread | | FIN Tweet vs. AVERAGE EMBI Spread | | Daily Bid–Ask vs. Daily EMBI Spread | |
|---|---|---|---|---|---|---|
|  | $R^2$ | Adjusted $R^2$ | $R^2$ | Adjusted $R^2$ | $R^2$ | Adjusted $R^2$ |
| Colombia | 0.003 | 0 | 0.015 | 0.012 | 0.001 | 0.001 |
| Ecuador | 0.003 | −0.001 | 0 | −0.004 | 0.089 | 0.088 |
| Peru | 0.055 | 0.052 | 0.055 | 0.052 | 0.013 | 0.012 |

* Bi-weekly accumulated impact of tweets; daily bid–ask; and EMBI spreads.

After discarding the larger dataset relationships with the dependent variable, further exploration of the quarterly macroeconomic data for each country contributed useful insights into the associations that had predictive attributes for the EMBI spread.

### 3.5. Colombia

I use 12 of the 14 independent variables in the regression analysis for Colombia since the two binary (dummy) variables, if it is traded or not and investment grade or not, are rendered useless when eliminating the corresponding missing data due to the remaining observations all having the same attribute. These 12 variables were regressed against all three variants of the EMBI spread with corresponding quarterly periods, and the average spread of the EMBI provided the highest adjusted $R^2$ of 0.9, a suspiciously high result that warranted further inspection of the model's parameters and VIF test for multicollinearity (VIF > 5). This analysis led to the elimination of unacceptably high VIF factors for five independent variables (Table 7).

**Table 7.** VIF Multicollinearity test for Colombia. Eliminated Variables.

|  | Colombia Endogenous Risk | Colombia STARTING A BUSINESS | Colombia CORRUPTION | Colombia Reserves/GDP | Colombia Bid Ask Spread (Liquidity) |
|---|---|---|---|---|---|
| Tolerance | 0.053 | 0.028 | 0.043 | 0.030 | 0.065 |
| VIF | 18.813 | 35.544 | 23.191 | 32.991 | 15.393 |

The surviving model had an adjusted $R^2$ of 0.71, the null hypothesis related to foreign exchange rate variables (FX DepR_AppR and STD. DEV of F/X) was rejected over the period of analysis (Table 8 and Figure 2).

**Table 8.** Model parameters for Colombia.

| Source | t | Pr > \|t\| | Accept H$_0$ | Reject H$_0$ |
|---|---|---|---|---|
| Colombia FX DepR_AppR | 4.647 | <0.0001 | | Hypothesis 3 |
| STD. DEV of F/X | 7.083 | <0.0001 | | Hypothesis 3 |
| Colombia Bid Ask Spread (Liquidity) | 1.721 | 0.098 | Hypothesis 3 | |
| Colombia AVERAGE PERIOD RATING | 0.953 | 0.350 | Hypothesis 3 | |

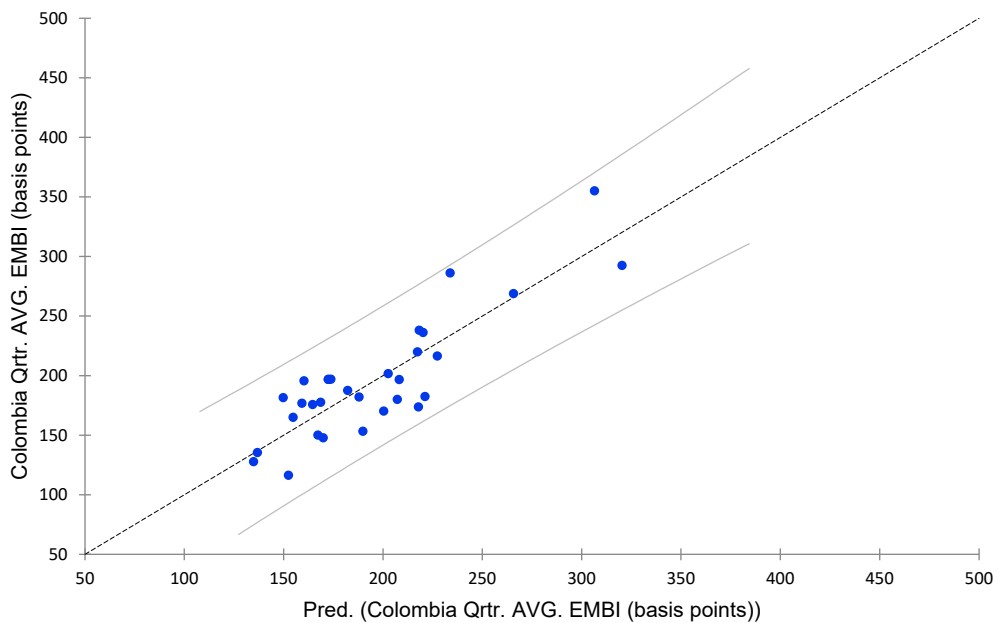

**Figure 2.** Colombia predicted vs. actual EMBI spread.

### 3.6. Ecuador

Two CRs were computed for the country of Ecuador. The first CR was discarded due to its scant number of observations (19) that resulted from the limited data of the bid–ask spread variable for the Ecuadorian sovereign issue and affected the surviving corresponding observations of the other independent variables.

The regression was then performed with 10 of the 14 independent variables after eliminating the two foreign exchange variables (as previously mentioned, Ecuador adopted the US dollar as its official currency at the turn of the millennium), the bid–ask spread variable, and finally, the investment grade-or-not binary variable, since, throughout the period of analysis, the country's debt never held an investment-grade credit rating.

Further inspection for multicollinearity eliminated the STARTING A BUSINESS (VIF = 9.8) and the CORRUPTION (VIF = 8.3) explanatory variables; the results of the significance of the remaining variables are displayed in Table 9 and Figure 3.

**Table 9.** Model parameters for Ecuador.

| Source | t | Pr > \|t\| | Accept H$_0$ | Reject H$_0$ |
|---|---|---|---|---|
| Ecuador Endogenous Risk | −3.294 | 0.002 | | Hypothesis 1 |
| Ecuador Debt Svce./GDP | 2.979 | 0.005 | | Hypothesis 4 |
| Ecuador FDI/GDP | 0.054 | 0.957 | Hypothesis 4 | |
| Ecuador Reserves/GDP | 1.569 | 0.126 | Hypothesis 4 | |
| Ecuador ALL Tweets | −1.661 | 0.106 | Hypothesis 5 | |
| Ecuador FIN Tweets | −1.894 | 0.067 | Hypothesis 5 | |
| Ecuador AVERAGE PERIOD RATING | −1.324 | 0.195 | Hypothesis 3 | |

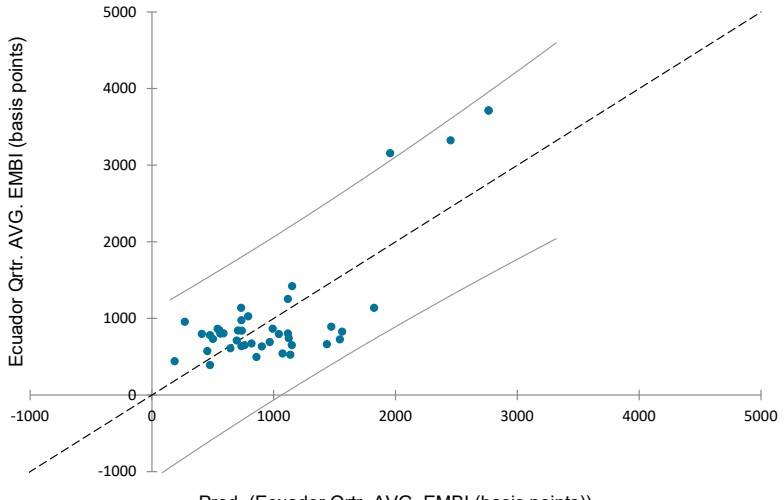

**Figure 3.** Ecuador predicted vs. actual EMBI spread.

### 3.7. Peru

Two sets of CRs were performed with the data for Peru. The first CR included the ALL tweet and FIN tweet variables, and the adjusted $R^2$ was low for all three variations of the EMBI spread dependent variable (0.23, 0.39, and 0.51). Furthermore, upon inspection for the presence of multicollinearity, eliminating STARTING A BUSINESS (VIF = 17.5), CORRUPTION (VIF = 28.9), and bid–ask spread (VIF = 4.8), the surviving variables and their resulting hypothesis tests and predicted plot are exhibited in Table 10 and Figure 4.

**Table 10.** Model parameters for Peru.

| Source | t | Pr > \|t\| | Accept $H_0$ | Reject $H_0$ |
|---|---|---|---|---|
| Peru Endogenous Risk | −0.541 | 0.593 | Hypothesis 1 | |
| Peru Debt Svce./GDP | −1.521 | 0.141 | Hypothesis 2 | |
| Peru Reserves/GDP | 1.431 | 0.165 | Hypothesis 2 | |
| Peru FX DepR_AppR | −1.111 | 0.277 | Hypothesis 3 | |
| STD. DEV of F/X | 3.070 | 0.005 | | Hypothesis 3 |
| Peru ALL Tweets | −0.308 | 0.761 | Hypothesis 5 | |
| Peru FIN Tweets | −1.599 | 0.123 | Hypothesis 5 | |
| Peru AVG. Invest. Grd. (1) Or Not (0) | −0.502 | 0.620 | Hypothesis 3 | |

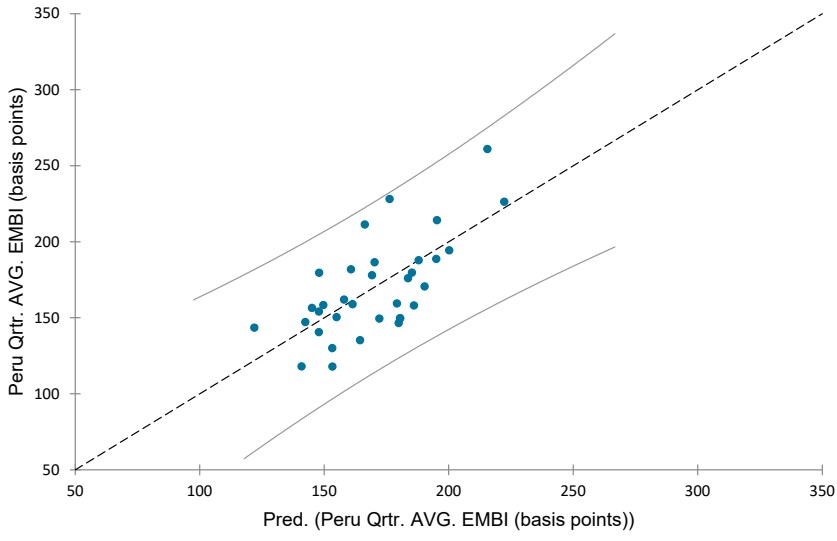

**Figure 4.** Peru predicted vs. actual EMBI spread.

## 4. Discussion and Conclusions

### 4.1. Discussion

In this study, I investigate the relationship between various types of independent variables and the resulting EMBI spread for three countries for the period ranging from as early as 2000 (Ecuador) to 2018.

The regression results of the independent variables related to the foreign exchange rates and policy confirm previous literature regarding its impact on country risk for the countries of Colombia and Peru. Namely, the standard deviation of the foreign exchange rate of the local currency had a positive impact on the EMBI spread. The larger the variability, the greater the EMBI spread of the sovereign bond issue. Furthermore, this result confirms the effects of active currency management to rein in credit spreads, to the extent that the foreign exchange quarterly standard deviation was the only significant variable to explain the country risk spread for the country of Peru.

The results from macroeconomic fundamentals have an impact on the EMBI spread for all three countries studied. This outcome is in line with previous research from Edwards (1984, 1985, 1986), Grandes (2007), Goldman Sachs (2000), and others.

The endogenous risk variable was only relevant for the country of Ecuador, and the resulting direction of impact was contrary to expectations. In characterizing an economy as a portfolio of assets, I anticipated that the larger endogenous risk (variability of economic activity) would result in a larger EMBI spread in the same way that a larger variability of an asset return in the capital markets demands a higher return.

However, the regression captured a negative relationship between these two variables. A plausible explanation, which warrants further research, may be that despite the higher variability of an economy causing a larger endogenous risk, there is a larger, more relevant impact related to economic growth (i.e., a variable not studied), which could be caused by a significant contribution from a particular economic segment, such as oil exploration or mining. The notion is that although this significant impact from a particular economic segment could cause a larger correlation with other economic activities in an economy and thus a larger covariance of economic activities, hence a larger endogenous risk, this impact would logically be conducive to the reduction of the EMBI spread as a result of the economic growth and this newly booming economic activity. Yet, this relationship between the jump in oil prices and the reduction in country risk has been demonstrated to be insignificant in oil-exporting countries, as reported by Bouri (2019).

### 4.2. Conclusions

Research involving the determinants of country risk falls under three categories: (1) macroeconomic fundamental variables, (2) sentiment variables, and (3) foreign exchange variables.

I posit an intrinsic risk measure for an economy which I call endogenous risk, and place it under the fundamental macroeconomic variable category. However, only Ecuador of the three countries studied provided a significant result from that variable. I use quarterly observations of economic activity by the industrial sector to perform a correlation and covariance measurement, which proxies the riskiness of an economy. This quarterly measurement has limitations for at least two reasons. Firstly, the release of the information does not coincide with the observation or measurement of the country's risk spread. Secondly, economic activity tabulation occurs quarterly, at best, yet the EMBI spread is observed daily.

The sentiment variables category, where I present a measure of impact from Twitter [®] activity, provided no significant results as a predictor of country risk. Research has clearly established the impact on securities' premiums or discounts resulting from price activity caused by sentiment or irrationality (T. L. D. Huynh et al. 2021). Contemporary effects of this nature are evidenced in the form of fake news and social media activity. What is also clear is that this effect has been more broadly researched for stocks and not bonds. Furthermore, the use of the average of a country risk measure over a period of time (average

spread of a trimester in this research) would logically tend to smooth out any impacts resulting from social media, thus rendering its significance useless.

Foreign exchange rate variables fall under the third category of research regarding their impact on country risk. Here the significance is clear for both countries, Colombia and Peru, whose currency is actively managed with the intent to fend off depreciation or appreciation and target inflation (Libman 2019). Appreciation and depreciation measurements of foreign exchange, as well as the standard deviation of the foreign exchange, are of significance to predict the EMBI spreads of both Colombia and Peru.

Further research regarding the posited variable of endogenous risk could involve a more detailed level of analysis of the International Standard Industrial Classification (ISIC) by economic activity. Furthermore, the release date of fundamental macroeconomic information treated as an event study that informs the level of EMBI spread could prove a useful focal point to determine the time lag of these regressors against the measure of country risk. Similarly, the Twitter® activity measure could be broadened or supplemented to include other proxies of sentiment such as Thomson Reuters MarketPsych® or RavenPack® (Gan et al. 2020).

**Funding:** This research received no external funding.

**Institutional Review Board Statement:** Not applicable.

**Informed Consent Statement:** Not applicable.

**Data Availability Statement:** The data presented in this study are available upon request from the corresponding author.

**Acknowledgments:** I would like to acknowledge José Luis Gallizo (joseluis.gallizo@udl.cat) for his support and recommendations regarding my research.

**Conflicts of Interest:** The author declares no conflict of interest.

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
