# Peer review of "Inferences from Portfolio Theory and Efficient Market Hypothesis to the Impact of Social Media on Sovereign Debt: Colombia, Ecuador, and Peru"

_jrfm, doi:10.3390/jrfm15040160_

Round 1

Reviewer 1 Report

This paper examined that the impact of EMBI for Latin America countries such as Columbia, Ecuador and Peru. The results of this paper can provide some marginal contributions, so that this paper can publish. However, there are several issues in this study that need to be addressed before it is ready to be published for a publication in a journal.

  • Summarize key empirical finding results in introduction with concise manner.
  • Please rewrite the abstract, because it is so confusing.
  • Please emphasize the novelty of this paper. In line 428, you said that “the impact of the variables related to the FX of the local currency v.s. the U$ on the EMBI spread.” However, this is proved already by many previous studies.
  • Please delete line 49- line 55, which is not relevant for this study.
  • Once again, please delete line 117-line 123.
  • Please rewrite Hypothesis 1 to Hypothesis 3. It is very confusing. Please keep clear and concise manner.
  • The author should provide robustness check of empirical results.
  • Please explain why the empirical results are different among 3 Latin American countries.
  • Please clarify the line 381, “Moreover, the DW statistic….” It is very confusing.

Author Response

Reviewer 1.

  • Summarize key empirical finding results in introduction with concise manner.

I have redacted as suggested

  • Please rewrite the abstract, because it is so confusing.

I have made it more concise and to the point.  Thank you for your suggestion:

For three countries of similar economic characteristics, I ratify previous studies of the impact of fundamental macroeconomic and foreign exchange variables having an effect on country risk, as captured by the Emerging Market Bond Index (EMBI).  I contribute to existing research, first by calculating a proxy of risk I call endogenous risk that analyzes the quarterly variability of economic activity, and second, by calculating a variable of sentiment from Twitter activity to gauge the impact of both on the country risk metric.  Foreign exchange variables are the most significant determinants of risk for the countries of Colombia and Peru who actively manage their currency, while Ecuador’s country risk is mostly affected by endogenous risk and macroeconomic fundamentals.

  • Please emphasize the novelty of this paper. In line 428, you said that “the impact of the variables related to the FX of the local currency v.s. the U$ on the EMBI spread.” However, this is proved already by many previous studies.

The variables contributed relate to the impact calculation of Twitter activity and what I call the Endogenous Risk by computing the variability of economic activity by ISIC (quarterly)

  • Please delete line 49- line 55, which is not relevant for this study.

I have redacted as suggested.

  • Once again, please delete line 117-line 123.

I have redacted as suggested.

  • Please rewrite Hypothesis 1 to Hypothesis 3. It is very confusing. Please keep clear and concise manner.

I have reorganized the Hypothesis and placed them in a more organized fashion, as suggested.

  • The author should provide robustness check of empirical results.

I have provided robustness checks for multicollinearity and unit root (ADF)

  • Please explain why the empirical results are different among 3 Latin American countries

The time frame selected for analysis (2000-2019) is a period immediately after Ecuador adopted the US Dollar as its official currency, resulting in an experimental setting conducive for research of these three geographically neighboring, but more importantly from a research question perspective, all three are commodity dependent countries with differing exchange rate policies ranging from substantially non-existent (as a result of Ecuador’s dollarization of its economy) to inflation targeting monetary policy and exchange rate management in Colombia and Peru. 

  • Please clarify the line 381, “Moreover, the DW statistic….” It is very confusing.

I have redacted as suggested and eliminated that portion

Reviewer 2 Report

The present form is not acceptable it requires major modification to improve the novelty.

Author Response

Reviewer 2

The present form is not acceptable it requires major modification to improve the novelty.

I have thoroughly expanded and redacted the article with the help of colleagues and reviewers suggestions and changes.  Please look over my revised article.  Thank you.

Reviewer 3 Report

Please attend the next mandatory suggestions:

  1. The author Didn't explain with are the three countries of interest selected. please, give a strong foundation to your selection.
  2. The statements of lines 34 to 41 about the pricing of risk-premiums i.e. EMBI indexes, need references. Otherwise is the author's mere opinion without support. please give your references (Academic research papers or history research books).
  3. The introduction section is too short. I should see the author's hypothetical position or hypothesis to be tested and why is the author doing what it is doing in the paper. Please, expand.
  4. The Sovereign debt markets are not as old as the 17th Century. The author must expand his historical perspective with other Financial markets references besides Ferguson. The Sovereign debt market dates back to the 14th Century with Florence. Amsterdam is the place of the first stock market. Please correct and expand your references.
  5. The previous literature review looks strong but it needs enhancements. There is a vague link between your work and the motivations it receives from the cited references. Also in that literature review, I can't see how your model variables are related (motivated) with these previous references and how does the author expands the literature with his work. Please, expand your references and relate them to your work.
  6. The link Twiter sentiment-Fixed income markets, Twitter sentiment-EMBI spreads are too weak (one reference). There is more work published in Fixed income markets or other security markets about the impact of social media sentiment. Please, expand your references and relate these works with yours as motivations for your tests.
  7. Section 3 "The data" looks more like a bullets list or a sort of list. you give general detail but you didn't say how did you process the input data. Please write this section as a description of your test for Scientific replication purposes.
  8. Also related to the previous suggestion, because you didn't detail your motivations in the literature review, there is no link or foundation of your list of variables. With due respect, your work is good but this list of variables looks more like an "I think that these variables look good in my test..." I no it is not the case of your paper, but you must be careful of how do you write it in journals of high projection and potential as JFRM. Please attend this.
  9. In section 3.4 I finally read the hypothesis. Please, keep in mind that, when someone reads your abstract, she or he reads the conclusions next and, if there's more interest in your paper, she or he reads the introduction section. Your introduction section sells poorly your work and it doesn't say "what are you doing, how are you doing it, what will our tests be and what do you expect from these". The abstract should say "what did you do, how did you do it, what are your main findings and contributions" and your conclusions should say the same of the abstract, enhanced with the guidelines for further research. These guidelines, given the limitations or opportunity areas that you find in your work. The literature review gives strength to your tests and variables selection. Please, enhance those sections as suggested.
  10. Your hypothesis could look better if they are of the type "There is a positive (or negative) relation between X and Y with Z" (general hypothesis) and some particular ones for each variable (X and Y). You also must highlight your theoretical position in the discussion of these hypotheses.
  11. You didn't specify clearly (or I missed it with a lot of information) the periodicity and length (start and end dates), and if you used the raw data, their original units, and if you used logs, log differences, etc. please, complete that description.
  12. You didn't mention if you made a unit root test. It is important because you are dealing with time series in an OLS multivariate regression. You are using the EMBI first differences. Why first differences and not log-differences that are less prone to heteroskedasticity than the former? please explain or make either the unit root tests, along with a strong explanation. In my personal experience (with due respect) It is more comfortable to work with log first differences. The scale is more homogeneous, the effect of unit roots disappears and your factor betas are interpreted as "elasticities" a Delta% change in X leads to a Delta% change in Y. This is even consistent with other factor models either in Classical Financial Economics (Sharpe, Fama, and Co.) or Behavioral Financial Economics that is the research area that your work belongs with the Twitter sentiment measure.
  13. In your literature review, please include some recent behavioral Financial Economics papers. If you depart from the "rational" equilibrium models in which market sentiment is not considered, you must discuss in your literature, previous works in this area, how do these motivate the inclusion of Twitter sentiment, and how do you expand the literature with your work.
  14. There is a potential drawback and potential risk in your tests: you didn't use robust standard error such as the ones of Newey-West. Also, you didn't explain if there is a risk of multicollinearity in your regressors. Please make the proper multicollinearity tests (if applicable) and robust standard error estimation. If your conclusions do not change, please explain in the answer letter with the two regression tables: your current ones and the robust estimation ones to see that really the p-values didn't change significantly and that your conclusions hold.
  15. Please make a brief corollary in your results section. This means that your results section must be organized as follows:
    1. 3.1 input data processing
    2. 3.3 Results discussion
    3. 3.4 Corollary of results and comparison among countries.
  16. In the conclusions section, please mention how are your results useful in Academia and for practitioners (also in the abstract that shouldn't be more than 180 words).
  17. In the same section, please mention the guidelines for further research. That is, how can someone improve your tests, or what are the next research tests to be made with your results. What improvements can be done to your research?
  18. Appendix A and B are not necessary. Those are for mathematical proofs or algorithms that your paper does not have. Those appendices must be in the data processing section. Also, section 4 of materials and methods is not necessary. Please check that with the Editor.
  19. These are some suggested works that could be included in your references:
  • http://review-rper.com/index.php/rper/article/view/69/215
  • https://www.emerald.com/insight/content/doi/10.1108/978-1-80071-070-220211009/full/html 
  • https://www.sciencedirect.com/science/article/pii/S0148296321008559 
  • https://link.springer.com/article/10.1007/s13278-020-00709-9 

Please, mark in blue your changes in the new paper's version.

Author Response

Reviewer 3

  1. The author Didn't explain with are the three countries of interest selected. please, give a strong foundation to your selection.

The time frame selected for analysis (2000-2019) is a period immediately after Ecuador adopted the US Dollar as its official currency, resulting in an experimental setting conducive for research of these three geographically neighboring, but more importantly from a research question perspective, all three are commodity dependent countries with differing exchange rate policies ranging from substantially non-existent (as a result of Ecuador’s dollarization of its economy) to inflation targeting monetary policy and exchange rate management in Colombia and Peru. 

  1. The statements of lines 34 to 41 about the pricing of risk-premiums i.e. EMBI indexes, need references. Otherwise is the author's mere opinion without support. please give your references (Academic research papers or history research books).

Morgan, J.P., 1999. Introducing the JP Morgan Emerging Markets Bond Index Global (EMBI Global). Methodology Brief (August).  Please review the expanded section which delves into the details. 

  1. The introduction section is too short. I should see the author's hypothetical position or hypothesis to be tested and why is the author doing what it is doing in the paper. Please, expand.

I have expanded the introduction and reorganized the section to include the hypotheses expectations.

  1. The Sovereign debt markets are not as old as the 17th Century. The author must expand his historical perspective with other Financial markets references besides Ferguson. The Sovereign debt market dates back to the 14th Century with Florence. Amsterdam is the place of the first stock market. Please correct and expand your references.

I believe I was misunderstood because of my convoluted writing (apologies).  I believe I have written it more clearly after your comment.  I must add that I am in effect referring to the fact that the debt market predates the stock market.  I have corrected and expanded my references.

  1. The previous literature review looks strong but it needs enhancements. There is a vague link between your work and the motivations it receives from the cited references. Also in that literature review, I can't see how your model variables are related (motivated) with these previous references and how does the author expands the literature with his work. Please, expand your references and relate them to your work.

I have included research and literature pertinent to market sentiment notions. 

  1. The link Twiter sentiment-Fixed income markets, Twitter sentiment-EMBI spreads are too weak (one reference). There is more work published in Fixed income markets or other security markets about the impact of social media sentiment. Please, expand your references and relate these works with yours as motivations for your tests.

I have included research and literature pertinent to market sentiment notions. 

  1. Section 3 "The data" looks more like a bullets list or a sort of list. you give general detail but you didn't say how did you process the input data. Please write this section as a description of your test for Scientific replication purposes.

I have expanded the explanation of my variable selection and used existing research to motivate the selection of my fourteen variables, specifically the selection of variables derives from the 3 categories of research as described by Mari del Cristo and Gómez-Puig (2017).

  1. Also related to the previous suggestion, because you didn't detail your motivations in the literature review, there is no link or foundation of your list of variables. With due respect, your work is good but this list of variables looks more like an "I think that these variables look good in my test..." I no it is not the case of your paper, but you must be careful of how do you write it in journals of high projection and potential as JFRM. Please attend this.

I have expanded the explanation of my variable selection and used existing research to motivate the selection of my fourteen variables, specifically the selection of variables derives from the 3 categories of research as described by Mari del Cristo and Gómez-Puig (2017).

  1. In section 3.4 I finally read the hypothesis. Please, keep in mind that, when someone reads your abstract, she or he reads the conclusions next and, if there's more interest in your paper, she or he reads the introduction section. Your introduction section sells poorly your work and it doesn't say "what are you doing, how are you doing it, what will our tests be and what do you expect from these". The abstract should say "what did you do, how did you do it, what are your main findings and contributions" and your conclusions should say the same of the abstract, enhanced with the guidelines for further research. These guidelines, given the limitations or opportunity areas that you find in your work. The literature review gives strength to your tests and variables selection. Please, enhance those sections as suggested.

I have rewritten the abstract and reorganized the hypothesis to be placed in the introduction.  I also have strengthened my introduction section and concluded it with an overview of the organization of the paper. 

  1. Your hypothesis could look better if they are of the type "There is a positive (or negative) relation between X and Y with Z" (general hypothesis) and some particular ones for each variable (X and Y). You also must highlight your theoretical position in the discussion of these hypotheses.

I have placed the hypothesis in a different section and believe this to be a better flow of the paper.

  1. You didn't specify clearly (or I missed it with a lot of information) the periodicity and length (start and end dates), and if you used the raw data, their original units, and if you used logs, log differences, etc. please, complete that description.

I have clarified the period selected in the introduction and in the data and methodology.

  1. You didn't mention if you made a unit root test. It is important because you are dealing with time series in an OLS multivariate regression. You are using the EMBI first differences. Why first differences and not log-differences that are less prone to heteroskedasticity than the former? please explain or make either the unit root tests, along with a strong explanation. In my personal experience (with due respect) It is more comfortable to work with log first differences. The scale is more homogeneous, the effect of unit roots disappears and your factor betas are interpreted as "elasticities" a Delta% change in X leads to a Delta% change in Y. This is even consistent with other factor models either in Classical Financial Economics (Sharpe, Fama, and Co.) or Behavioral Financial Economics that is the research area that your work belongs with the Twitter sentiment measure.

You make a great point about the logs (no offense taken).  I did do log differences but precisely for concerns of spurious correlation, I had also performed the following for the dependent: (1) a mean (expected value) of EMBI spread for the quarterly period, and (2) a standard deviation of the EMBI spread for the quarterly period.  The spread of daily observations averaged (expected value of the quarter) over the quarterly period were stationary for the dependent of Ecuador Peru and Colombia. In any event, I have performed logs and log differences and ran an ADF test to check for non-stationarity and have kept the Average EMBI Spread over the quarter as it is a stationary variable.

Below is the output from ADF for all relevant Ys.

ADF Test-Colombia Qrtr. AVG. EMBIG (basis points)

criteria

aic

drift

yes

trend

yes

lag

2

alpha

0.05

tau-stat

-3.767121037

tau-crit

-3.485755462

stationary

yes

aic

10.78909986

bic

10.93643202

lags

1

coeff

-0.350564093

p-value

0.025746664

ADF Test - Ecuador Qrtr. AVG. EMBIG (basis points)

criteria

aic

drift

yes

trend

yes

lag

2

alpha

0.05

tau-stat

-6.404796418

tau-crit

-3.465303031

stationary

yes

aic

14.691983

bic

14.81652706

lags

1

coeff

-0.432500423

p-value

< .01

ADF Test-Peru Qrtr. AVG. EMBIG (basis points)

criteria

aic

drift

yes

trend

yes

lag

2

alpha

0.05

-4.165290363

tau-crit

-3.498968825

stationary

yes

aic

10.30898414

bic

10.46799643

lags

1

coeff

-0.400702977

p-value

< .01

  1. In your literature review, please include some recent behavioral Financial Economics papers. If you depart from the "rational" equilibrium models in which market sentiment is not considered, you must discuss in your literature, previous works in this area, how do these motivate the inclusion of Twitter sentiment, and how do you expand the literature with your work.

I have included research and literature pertinent to market sentiment notions. 

  1. There is a potential drawback and potential risk in your tests: you didn't use robust standard error such as the ones of Newey-West. Also, you didn't explain if there is a risk of multicollinearity in your regressors. Please make the proper multicollinearity tests (if applicable) and robust standard error estimation. If your conclusions do not change, please explain in the answer letter with the two regression tables: your current ones and the robust estimation ones to see that really the p-values didn't change significantly and that your conclusions hold.

Thank you for your valuable suggestion.  This changed my conclusions about the significant variables and I have redacted the section in the manuscript accordingly.

I used VIF (Variance Inflation Factor) to discard independent variables which exhibited unacceptable multicollinearity levels (VIF>5).  Below are the VIF tables for the Complete Regression (the first regressions ran using all independent variables possible) and below the regressions finally selected based on acceptable VIFs.

Colombia: Multicolinearity statistics (before removal of VIF>5):

Endogenous Risk

Colombia STARTING A BUSINESS

Colombia CORRUPTION

Colombia FDI/GDP

Colombia Debt Svce./GDP

Colombia Reserves/GDP

Colombia FX DepR_AppR

STD. DEV of F/X

Colombia ALL Tweets

Colombia FIN Tweets

Colombia Bid Ask Spread (Liquidity)

Colombia AVERAGE PERIOD RATING

Tolerance

0.053

0.028

0.043

0.228

0.088

0.030

0.328

0.279

0.426

0.334

0.065

0.351

VIF

18.813

35.544

23.191

4.394

11.372

32.991

3.052

3.589

2.345

2.998

15.393

2.847

Colombia: Surviving Variables:

Colombia FX DepR_AppR

STD. DEV of F/X

Colombia Bid Ask Spread (Liquidity)

Colombia AVERAGE PERIOD RATING

Tolerance

0.690

0.538

0.762

0.583

VIF

1.449

1.858

1.313

1.717

Source

t

Pr > |t|

Colombia FX DepR_AppR

4.647

<0.0001

STD. DEV of F/X

7.083

<0.0001

Colombia Bid Ask Spread (Liquidity)

1.721

0.098

Colombia AVERAGE PERIOD RATING

0.953

0.350

Ecuador: Multicolinearity statistics (before removal of VIF>5):

Ecuador Endogenous Risk

Ecuador STARTING A BUSINESS

Ecuador CORRUPTION

Ecuador FDI/GDP

Ecuador Debt Svce./GDP

Ecuador Reserves/GDP

Ecuador ALL Tweets

Ecuador FIN Tweets

Ecuador AVERAGE PERIOD RATING

Tolerance

0.461

0.102

0.120

0.410

0.324

0.310

0.742

0.454

0.323

VIF

2.169

9.831

8.335

2.437

3.085

3.223

1.348

2.204

3.096

Ecuador Endogenous Risk

Ecuador FDI/GDP

Ecuador Debt Svce./GDP

Ecuador Reserves/GDP

Ecuador ALL Tweets

Ecuador FIN Tweets

Ecuador AVERAGE PERIOD RATING

Tolerance

0.622

0.596

0.410

0.337

0.834

0.649

0.586

VIF

1.609

1.677

2.442

2.968

1.199

1.541

1.706

Ecuador: Surviving Variables:

Source

t

Pr > |t|

Ecuador Endogenous Risk

-3.294

0.002

Ecuador FDI/GDP

0.054

0.957

Ecuador Debt Svce./GDP

2.979

0.005

Ecuador Reserves/GDP

1.569

0.126

Ecuador ALL Tweets

-1.661

0.106

Ecuador FIN Tweets

-1.894

0.067

Ecuador AVERAGE PERIOD RATING

-1.324

0.195

Peru: Multicolinearity statistics (before removal of VIF>5):

Peru Endogenous Risk

Peru STARTING A BUSINESS

Peru CORRUPTION

Peru FDI/GDP

Peru Debt Svce./GDP

Peru Reserves/GDP

Peru FX DepR_AppR

STD. DEV of F/X

Peru ALL Tweets

Peru FIN Tweets

Peru Bid Ask Spread (Liquidity)

Peru AVERAGE PERIOD RATING

Peru AVG. Invest. Grd. (1) Or Not (0)-0

Peru AVG. Invest. Grd. (1) Or Not (0)-1

Tolerance

0.600

0.057

0.035

0.214

0.318

0.149

0.501

0.576

0.253

0.556

0.209

0.067

0.671

0.671

VIF

1.666

17.495

28.860

4.667

3.144

6.701

1.996

1.736

3.958

1.798

4.794

14.940

1.490

1.490

Peru: Surviving Variables

Peru Endogenous Risk

Peru Debt Svce./GDP

Peru Reserves/GDP

Peru FX DepR_AppR

STD. DEV of F/X

Peru ALL Tweets

Peru FIN Tweets

Peru AVG. Invest. Grd. (1) Or Not (0)

Tolerance

0.666

0.823

0.561

0.691

0.802

0.472

0.658

0.874

VIF

1.502

1.215

1.784

1.446

1.247

2.119

1.520

1.144

Source

t

Pr > |t|

Peru Endogenous Risk

-0.541

0.593

Peru Debt Svce./GDP

-1.521

0.141

Peru Reserves/GDP

1.431

0.165

Peru FX DepR_AppR

-1.111

0.277

STD. DEV of F/X

3.070

0.005

Peru ALL Tweets

-0.308

0.761

Peru FIN Tweets

-1.599

0.123

Peru AVG. Invest. Grd. (1) Or Not (0)

-0.502

0.620

  1. Please make a brief corollary in your results section. This means that your results section must be organized as follows:
    1. 3.1 input data processing
    2. 3.3 Results discussion
    3. 3.4 Corollary of results and comparison among countries.

I have expanded the results to include the suggested organization.

  1. In the conclusions section, please mention how are your results useful in Academia and for practitioners (also in the abstract that shouldn't be more than 180 words).

I have added the suggested redaction.

  1. In the same section, please mention the guidelines for further research. That is, how can someone improve your tests, or what are the next research tests to be made with your results. What improvements can be done to your research?

I have added the suggested redaction in the Conclusions section.

  1. Appendix A and B are not necessary. Those are for mathematical proofs or algorithms that your paper does not have. Those appendices must be in the data processing section. Also, section 4 of materials and methods is not necessary. Please check that with the Editor.

I have redacted and included this information in the data processing section.       

  1. These are some suggested works that could be included in your references:

Thank you for your suggested review of literature, I have used two of them and added others which came form the references of the works you recommend. 

  • http://review-rper.com/index.php/rper/article/view/69/215
  • https://www.emerald.com/insight/content/doi/10.1108/978-1-80071-070-220211009/full/html 
  • https://www.sciencedirect.com/science/article/pii/S0148296321008559 
  • https://link.springer.com/article/10.1007/s13278-020-00709-9 

Please, mark in blue your changes in the new paper's version.

I have marked additions and your comments in blue and the sections moved to data in green.

Reviewer 4 Report

The literature review and theoretical derivation sections of this paper are very weak. This article could only be published if it were supplemented by a large number of up-to-date references and the logic of the first three sections were redesigned.

Author Response

Reviewer 4

The literature review and theoretical derivation sections of this paper are very weak. This article could only be published if it were supplemented by a large number of up-to-date references and the logic of the first three sections were redesigned.

I have thoroughly expanded and redacted the article with the help of colleagues and reviewers suggestions and changes.  Please look over my revised article.  Thank you.

my revised article.  Thank you.

Round 2

Reviewer 1 Report

In revision, yes you provided VIF for Columbia. However, you should provide VIF for Ecuador and Peru as well.

Author Response

Date of this review

21 Mar 2022 02:13:39

Comments and Suggestions for Authors

In revision, yes you provided VIF for Columbia. However, you should provide VIF for Ecuador and Peru as well.

I have redacted accordingly and included Tables with VIF and Tolerance for the countries of Ecuador and Peru as well. 

Reviewer 3 Report

The paper has increased its quality significantly. Still, there should be a mistake or an issue in the paper's new version in the editing system (susy). I don't see the blue text and green that you mention in your letter. It makes me a little bit difficult to check the changes made. 

Also, the citing format is not the journal's one. Please, refer to the author guidelines section and change the citing format. Mendeley or Endnote could be of great help for this purpose if you are not using them.

Please, send that paper's new version that you mention in your letter and attend the last recommendation of suggested works in full.

Author Response

Comments and Suggestions for Authors

The paper has increased its quality significantly. Still, there should be a mistake or an issue in the paper's new version in the editing system (susy). I don't see the blue text and green that you mention in your letter. It makes me a little bit difficult to check the changes made. 

I will include the previous revision version with Tracked changes in the additional documents section. Please let me know if you are able to see it.

Also, the citing format is not the journal's one. Please, refer to the author guidelines section and change the citing format. Mendeley or Endnote could be of great help for this purpose if you are not using them.

I consulted with editor and they suggested that format (it was also strange to me) with Chicago 17th with those parentheses at the beginning of the reference.

Please, send that paper's new version that you mention in your letter and attend the last recommendation of suggested works in full.

I have addressed it in full (please read below).  Please let me know if this is sufficient.

Submission Date

11 February 2022

Date of this review

18 Mar 2022 00:45:15

Reviewer 3 suggested: 

These are some suggested works that could be included in your references:

Thank you for your suggested review of literature, I have used two of them and added others which came form the references of the works you recommend. 

  • http://review-rper.com/index.php/rper/article/view/69/215

I was able to read: Oscar, V, Dora Aguilasocho-Montoya, Leticia Bollain-Parra, and Amador Durán-Sánchez. 2022. "The impact of COVID-19 news and investor sentiment in European stock pricing, a regional, country, and economic sector review." RPER (60): 165-177, however found works related to bond (fixed income/debt instruments to be more relevant to my article)

  • https://www.emerald.com/insight/content/doi/10.1108/978-1-80071-070-220211009/full/html 

I have read the abstract and article and decided to use a reference from work referred by your review below.

  • https://www.sciencedirect.com/science/article/pii/S0148296321008559 

From this article I read I preferred to use the following article from the references of this work: Gan, Baoqing, Vitali Alexeev, Ron Bird, and Danny Yeung. 2020. "Sensitivity to sentiment: News vs social media." International Review of Financial Analysis 67: 101390. https://doi.org/10.1016/j.irfa.2019.101390. https://dx.doi.org/10.1016/j.irfa.2019.101390.

  • https://link.springer.com/article/10.1007/s13278-020-00709-9 

Very interesting article about the term financial risk and the associations for data mining, but was satisfied with other selected works and included in the references.  Did not use this in the course of my description /redaction so was not able to cited.

Reviewer 4 Report

I have no further comments.

Author Response

Thank you for your review and improvements. All comments have been addressed.

This manuscript is a resubmission of an earlier submission. The following is a list of the peer review reports and author responses from that submission.